# VariBAD: A Very Good Method for Bayes-Adaptive Deep RL via Meta-Learning

**Luisa Zintgraf** [*]
University of Oxford

**Kyriacos Shiarlis**
Latent Logic

**Maximilian Igl**
University of Oxford

**Sebastian Schulze**
University of Oxford

**Yarin Gal**
University of Oxford

**Katja Hofmann**
Microsoft Research

**Shimon Whiteson**
University of Oxford

## ABSTRACT

Trading off exploration and exploitation in an unknown environment is key to maximising expected return during learning. A Bayes-optimal policy, which does so optimally, conditions its actions not only on the environment state but on the agent's uncertainty about the environment. Computing a Bayes-optimal policy is however intractable for all but the smallest tasks. In this paper, we introduce variational Bayes-Adaptive Deep RL (variBAD), a way to meta-learn to perform approximate inference in an unknown environment, and incorporate task uncertainty directly during action selection. In a grid-world domain, we illustrate how variBAD performs structured online exploration as a function of task uncertainty. We further evaluate variBAD on MuJoCo domains widely used in meta-RL and show that it achieves higher online return than existing methods.

## 1 INTRODUCTION

Reinforcement learning (RL) is typically concerned with finding an *optimal policy* that maximises expected return for a given Markov decision process (MDP) with an unknown reward and transition function. If these were known, the optimal policy could in theory be computed without environment interactions. By contrast, learning in an *unknown* environment usually requires trading off exploration (learning about the environment) and exploitation (taking promising actions). Balancing this trade-off is key to maximising expected return *during learning*, which is desirable in many settings, particularly in high-stakes real-world applications like healthcare and education (Liu et al., 2014; Yauney & Shah, 2018). A *Bayes-optimal policy*, which does this trade-off optimally, conditions actions not only on the environment state but on the agent's own uncertainty about the current MDP.

In principle, a Bayes-optimal policy can be computed using the framework of Bayes-adaptive Markov decision processes (BAMDPs) (Martin, 1967; Duff & Barto, 2002), in which the agent maintains a belief distribution over possible environments. Augmenting the state space of the underlying MDP with this belief yields a BAMDP, a special case of a belief MDP (Kaelbling et al., 1998). A Bayes-optimal agent maximises expected return in the BAMDP by systematically seeking out the data needed to quickly reduce uncertainty, but only insofar as doing so helps maximise expected return. Its performance is bounded from above by the optimal policy for the given MDP, which does not need to take exploratory actions but requires prior knowledge about the MDP to compute.

Unfortunately, planning in a BAMDP, i.e., computing a Bayes-optimal policy that conditions on the augmented state, is intractable for all but the smallest tasks. A common shortcut is to rely instead on *posterior sampling* (Thompson, 1933; Strens, 2000; Osband et al., 2013). Here, the agent periodically samples a single hypothesis MDP (e.g., at the beginning of an episode) from its posterior, and the policy that is optimal *for the sampled MDP* is followed until the next sample is drawn. Planning is far more tractable since it is done on a regular MDP, not a BAMDP. However, posterior sampling's exploration can be highly inefficient and far from Bayes-optimal.

---

[*]luisa.zintgraf@cs.ox.ac.uk

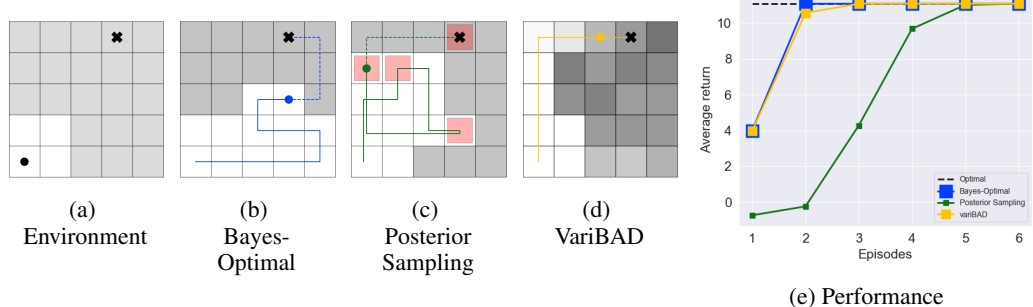

Figure 1: Illustration of different exploration strategies. **(a)** Environment: The agent starts at the bottom left and has to navigate to an unknown goal, located in the grey area. **(b)** A Bayes-optimal exploration strategy that systematically searches possible grid cells to find the goal, shown in solid (past actions) and dashed (future actions) blue lines. A simplified posterior is shown in the background in grey ($p = 1/$(number of possible goal positions left) of containing the goal) and white ($p = 0$). **(c)** Posterior sampling, which repeatedly samples a possible goal position (red squares) from the current posterior, takes the shortest route there, and updates its posterior. **(d)** Exploration strategy learned by variBAD. The grey background represents the approximate posterior the agent has learned. **(e)** Average return over all possible environments, over six episodes with 15 steps each (after which the agent is reset to the starting position). VariBAD results are averaged across 20 random seeds. The performance of any exploration strategy is bounded above by the optimal behaviour (of a policy with access to the true goal position). The Bayes-optimal agent matches this behaviour from the second episode, whereas posterior sampling needs six rollouts. VariBAD closely approximates Bayes-optimal behaviour in this environment.

Consider the example of a gridworld in Figure 1, where the agent must navigate to an *unknown* goal located in the grey area (1a). To maintain a posterior, the agent can uniformly assign non-zero probability to cells where the goal could be, and zero to all other cells. A Bayes-optimal strategy strategically searches the set of goal positions that the posterior considers possible, until the goal is found (1b). Posterior sampling by contrast samples a possible goal position, takes the shortest route there, and then resamples a different goal position from the updated posterior (1c). Doing so is much less efficient since the agent's uncertainty is not reduced optimally (e.g., states are revisited).

As this example illustrates, Bayes-optimal policies can explore much more efficiently than posterior sampling. A key challenge is to learn approximately Bayes-optimal policies while retaining the tractability of posterior sampling. In addition, the inference involved in maintaining a posterior belief, needed even for posterior sampling, may itself be intractable.

In this paper, we combine ideas from Bayesian RL, approximate variational inference, and meta-learning to tackle these challenges, and equip an agent with the ability to strategically explore unseen (but related) environments for a given distribution, in order to maximise its expected online return.

More specifically, we propose *variational Bayes-Adaptive Deep RL* (variBAD), a way to meta-learn to perform approximate inference on an unknown task,[1] and incorporate task uncertainty directly during action selection. Given a distribution over MDPs $p(M)$, we represent a single MDP $M$ using a learned, low-dimensional stochastic latent variable $m$ and jointly meta-train:

1. A variational auto-encoder that can infer the posterior distribution over $m$ in a new task, given the agent's experience, while interacting with the environment, and

2. A policy that conditions on this posterior belief over MDP embeddings, and thus learns how to trade off exploration and exploitation when selecting actions *under task uncertainty*.

Figure 1e shows the performance of our method versus the hard-coded optimal (with privileged goal information), Bayes-optimal, and posterior sampling exploration strategies. VariBAD's performance closely matches the Bayes-optimal one, matching optimal performance from the third rollout.

---

[1]We use the terms environment, task, and MDP, interchangeably.

Previous approaches to BAMDPs are only tractable in environments with small action and state spaces or rely on privileged information about the task during training. VariBAD offers a tractable and flexible approach for learning Bayes-adaptive policies tailored to the training task distribution, with the only assumption that such a distribution is available for meta-training. We evaluate our approach on the gridworld shown above and on MuJoCo domains that are widely used in meta-RL, and show that variBAD exhibits superior exploratory behaviour at test time compared to existing meta-learning methods, achieving higher returns during learning. As such, variBAD opens a path to tractable approximate Bayes-optimal exploration for deep reinforcement learning.

## 2 BACKGROUND

We define a Markov decision process (MDP) as a tuple $M = (\mathcal{S}, \mathcal{A}, R, T, T_0, \gamma, H)$ with $\mathcal{S}$ a set of states, $\mathcal{A}$ a set of actions, $R(r_{t+1}|s_t, a_t, s_{t+1})$ a reward function, $T(s_{t+1}|s_t, a_t)$ a transition function, $T_0(s_0)$ an initial state distribution, $\gamma$ a discount factor, and $H$ the horizon. In the standard RL setting, we want to learn a policy $\pi$ that maximises $\mathcal{J}(\pi) = \mathbb{E}_{T_0, T, \pi}\left[\sum_{t=0}^{H-1} \gamma^t R(r_{t+1}|s_t, a_t, s_{t+1})\right]$, the expected return. Here, we consider a multi-task meta-learning setting, which we introduce next.

### 2.1 TRAINING SETUP

We adopt the standard meta-learning setting where we have a distribution $p(M)$ over MDPs from which we can sample during meta-training, with an MDP $M_i \sim p(M)$ defined by a tuple $M_i = (\mathcal{S}, \mathcal{A}, R_i, T_i, T_{i,0}, \gamma, H)$. Across tasks, the reward and transition functions vary but share some structure. The index $i$ represents an unknown task description (e.g., a goal position or natural language instruction) or task ID. Sampling an MDP from $p(M)$ is typically done by sampling a reward and transition function from a distribution $p(R, T)$. During meta-training, batches of tasks are repeatedly sampled, and a small training procedure is performed on each of them, with the goal of learning to learn (for an overview of existing methods see Sec 4). At meta-test time, the agent is evaluated based on the average return it achieves *during learning*, for tasks drawn from $p$. Doing this well requires at least two things: (1) incorporating prior knowledge obtained in related tasks, and (2) reasoning about task uncertainty when selecting actions to trade off exploration and exploitation. In the following, we combine ideas from meta-learning and Bayesian RL to tackle these challenges.

### 2.2 BAYESIAN REINFORCEMENT LEARNING

When the MDP is unknown, optimal decision making has to trade off exploration and exploitation when selecting actions. In principle, this can be done by taking a Bayesian approach to reinforcement learning formalised as a Bayes-Adaptive MDP (BAMDP), the solution to which is a Bayes-optimal policy (Bellman, 1956; Duff & Barto, 2002; Ghavamzadeh et al., 2015).

In the Bayesian formulation of RL, we assume that the transition and reward functions are distributed according to a prior $b_0 = p(R, T)$. Since the agent does not have access to the true reward and transition function, it can maintain a belief $b_t(R, T) = p(R, T|\tau_{:t})$, which is the posterior over the MDP given the agent's experience $\tau_{:t} = \{s_0, a_0, r_1, s_1, a_1, \ldots, s_t\}$ up until the current timestep. This is often done by maintaining a distribution over the model parameters.

To allow the agent to incorporate the task uncertainty into its decision-making, this belief can be augmented to the state, resulting in hyper-states $s_t^+ \in \mathcal{S}^+ = \mathcal{S} \times \mathcal{B}$, where $\mathcal{B}$ is the belief space. These transition according to

$$
\begin{aligned}
T^+(s_{t+1}^+|s_t^+, a_t, r_t) &= T^+(s_{t+1}, b_{t+1}|s_t, a_t, r_t, b_t) \\
&= T^+(s_{t+1}|s_t, a_t, b_t)\, T^+(b_{t+1}|s_t, a_t, r_t, b_t, s_{t+1}) \\
&= \mathbb{E}_{b_t}\left[T(s_{t+1}|s_t, a_t)\right]\, \delta(b_{t+1} = p(R, T|\tau_{:t+1}))
\end{aligned}
\tag{1}
$$

i.e., the new environment state $s_t$ is the expected new state w.r.t. the current posterior distribution of the transition function, and the belief is updated deterministically according to Bayes rule. The reward function on hyper-states is defined as the expected reward under the current posterior (after the state transition) over reward functions,

$$
R^+(s_t^+, a_t, s_{t+1}^+) = R^+(s_t, b_t, a_t, s_{t+1}, b_{t+1}) = \mathbb{E}_{b_{t+1}}\left[R(s_t, a_t, s_{t+1})\right].
\tag{2}
$$

This results in a BAMDP $M^+ = (\mathcal{S}^+, \mathcal{A}, R^+, T^+, T_0^+, \gamma, H^+)$ (Duff & Barto, 2002), which is a special case of a belief MDP, i.e, the MDP formed by taking the posterior beliefs maintained by an agent in a partially observable MDP and reinterpreting them as Markov states (Cassandra et al., 1994). In an arbitrary belief MDP, the belief is over a hidden state that can change over time. In a BAMDP, the belief is over the transition and reward functions, which are constant for a given task. The agent's objective is now to maximise the expected return in the BAMDP,

$$\mathcal{J}^+(\pi) = \mathbb{E}_{b_0, T_0^+, T^+, \pi} \left[ \sum_{t=0}^{H^+ - 1} \gamma^t R^+(r_{t+1}|s_t^+, a_t, s_{t+1}^+) \right], \tag{3}$$

i.e., maximise the expected return in an initially unknown environment, while learning, within the horizon $H^+$. Note the distinction between the MDP horizon $H$ and BAMDP horizon $H^+$. Although they often coincide, we might instead want the agent to act Bayes-optimal within the first $N$ MDP episodes, so $H^+ = N \times H$. Trading off exploration and exploitation optimally depends heavily on how much time the agent has left (e.g., to decide whether information-seeking actions are worth it).

The objective in (3) is maximised by the Bayes-optimal policy, which automatically trades off exploration and exploitation: it takes exploratory actions to reduce its task uncertainty *only insofar as it helps to maximise the expected return within the horizon*. The BAMDP framework is powerful because it provides a principled way of formulating Bayes-optimal behaviour. However, solving the BAMDP is hopelessly intractable for most interesting problems.

The main challenges are as follows.

- We typically do not know the parameterisation of the true reward and/or transition model,
- The belief update (computing the posterior $p(R, T|\tau_{:t})$) is often intractable, and
- Even with the correct posterior, planning in belief space is typically intractable.

In the following, we propose a method that simultaneously meta-learns the reward and transition functions, how to perform inference in an unknown MDP, and how to use the belief to maximise expected online return. Since the Bayes-adaptive policy is learned end-to-end with the inference framework, no planning is necessary at test time. We make minimal assumptions (no privileged task information is required during training), resulting in a highly flexible and scalable approach to Bayes-adaptive Deep RL.

## 3 BAYES-ADAPTIVE DEEP RL VIA META-LEARNING

In this section, we present variBAD, and describe how we tackle the challenges outlined above. We start by describing how to represent reward and transition functions, and (posterior) distributions over these. We then consider how to meta-learn to perform approximate variational inference in a given task, and finally put all the pieces together to form our training objective.

In the typical meta-learning setting, the reward and transition functions that are unique to each MDP are unknown, but also share some structure across the MDPs $M_i$ in $p(M)$. We know that there exists a true $i$ which represents either a task description or task ID, but we do not have access to this information. We therefore represent this value using a learned stochastic latent variable $m_i$. For a given MDP $M_i$ we can then write

$$R_i(r_{t+1}|s_t, a_t, s_{t+1}) \approx R(r_{t+1}|s_t, a_t, s_{t+1}; m_i), \tag{4}$$
$$T_i(s_{t+1}|s_t, a_t) \approx T(s_{t+1}|s_t, a_t; m_i), \tag{5}$$

where $R$ and $T$ are shared across tasks. Since we do not have access to the true task description or ID, we need to *infer* $m_i$ given the agent's experience up to time step $t$ collected in $M_i$,

$$\tau_{:t}^{(i)} = (s_0, a_0, r_1, s_1, a_1, r_2, \ldots, s_{t-1}, a_{t-1}, r_t, s_t), \tag{6}$$

i.e., we want to infer the posterior distribution $p(m_i|\tau_{:t}^{(i)})$ over $m_i$ given $\tau_{:t}^{(i)}$ (from now on, we drop the sub- and superscript $i$ for ease of notation).

Recall that our goal is to *learn a distribution over the MDPs, and given a posteriori knowledge of the environment compute the optimal action*. Given the above reformulation, it is now sufficient to

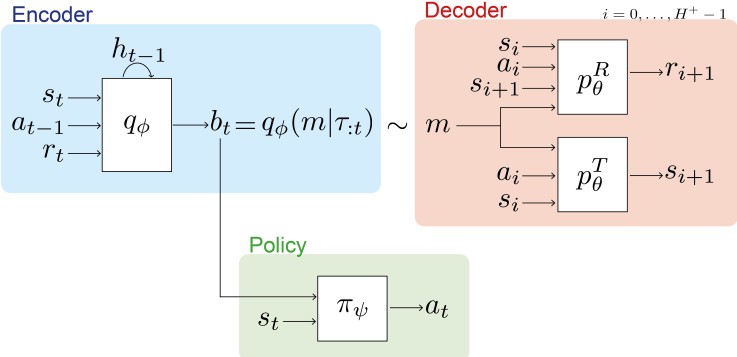

Figure 2: VariBAD architecture: A trajectory of states, actions and rewards is processed online using an RNN to produce the posterior over task embeddings, $q_\phi(m|\tau_{:t})$. The posterior is trained using a decoder which attempts to predict past and future states and rewards from current states and actions. The policy conditions on the posterior in order to act in the environment and is trained using RL.

reason about the embedding $m$, instead of the transition and reward dynamics. This is particularly useful when deploying deep learning strategies, where the reward and transition function can consist of millions of parameters, but the embedding $m$ can be a small vector.

### 3.1 APPROXIMATE INFERENCE

Computing the exact posterior is typically not possible: we do not have access to the MDP (and hence the transition and reward function), and marginalising over tasks is computationally infeasible. Consequently, we need to learn a model of the environment $p_\theta(\tau_{:H^+}|a_{:H^+-1})$, parameterised by $\theta$, together with an amortised inference network $q_\phi(m|\tau_{:t})$, parameterised by $\phi$, which allows fast inference at runtime *at each timestep* $t$. The action-selection policy is not part of the MDP, so an environmental model can only give rise to a distribution of trajectories when conditioned on actions, which we typically draw from our current policy, $a \sim \pi$. At any given time step $t$, our model learning objective is thus to maximise

$$\mathbb{E}_{\rho(M,\tau_{:H^+})}\left[\log p_\theta(\tau_{:H^+}|a_{:H^+-1})\right], \tag{7}$$

where $\rho(M,\tau_{:H^+})$ is the trajectory distribution induced by our policy and we slightly abuse notation by denoting by $\tau$ the state-reward trajectories, excluding the actions. In the following, we drop the conditioning on $a_{:H^+-1}$ to simplify notation.

Instead of optimising (7), which is intractable, we can optimise a tractable lower bound, defined with a learned approximate posterior $q_\phi(m|\tau_{:t})$ which can be estimated by Monte Carlo sampling (for the full derivation see AppendixA):

$$\mathbb{E}_{\rho(M,\tau_{:H^+})}\left[\log p_\theta(\tau_{:H^+})\right] \geq \mathbb{E}_\rho\left[\mathbb{E}_{q_\phi(m|\tau_{:t})}\left[\log p_\theta(\tau_{:H^+}|m)\right] - KL(q_\phi(m|\tau_{:t})||p_\theta(m))\right] \tag{8}$$
$$= ELBO_t.$$

The term $\mathbb{E}_q[\log p(\tau_{:H^+}|m)]$ is often referred to as the reconstruction loss, and $p(\tau_{:t}|m)$ as the decoder. The term $KL(q(m|\tau_{:t})||p_\theta(m))$ is the KL-divergence between our variational posterior $q_\phi$ and the prior over the embeddings $p_\theta(m)$. We set the prior to our previous posterior, $q_\phi(m|\tau_{:t-1})$, with initial prior $q_\phi(m) = \mathcal{N}(0, I)$.

As can be seen in Equation (8) and Figure 2, when the agent is at timestep $t$, we encode the *past trajectory* $\tau_{:t}$ to get the current posterior $q(m|\tau_{:t})$ since this is all the information available to perform inference about the current task. We then decode the entire trajectory $\tau_{:H^+}$ *including the future*, i.e., model $\mathbb{E}_q[p(\tau_{:H^+}|m)]$. This is different than the conventional VAE setup (and possible since we have access to this information during training). Decoding not only the past but also the future is important because this way, variBAD learns to perform inference about unseen states given the past.

The reconstruction term $\log p(\tau_{:H^+}|m)$ factorises as

$$\log p(\tau_{:H^+}|m, a_{:H^+-1}) = \log p((s_0, r_0, \ldots, s_{t-1}, r_{t-1}, s_t)|m, a_{:H^+-1}) \tag{9}$$

$$= \log p(s_0|m) + \sum_{i=0}^{H^+-1} \left[ \log p(s_{i+1}|s_i, a_i, m) + \log p(r_{i+1}|s_i, a_i, s_{i+1}, m) \right].$$

Here, $p(s_0|m)$ is the initial state distribution $T'_0$, $p(s_{i+1}|s_i, a_i; m)$ the transition function $T'$, and $p(r_{i+1}|s_t, a_t, s_{i+1}; m)$ the reward function $R'$. From now, we include $T'_0$ in $T'$ for ease of notation.

## 3.2 Training Objective

We can now formulate a training objective for learning the approximate posterior distribution over task embeddings, the policy, and the generalised reward and transition functions $R'$ and $T'$. We use deep neural networks to represent the individual components. These are:

1. The encoder $q_\phi(m|\tau_{:t})$, parameterised by $\phi$;

2. An approximate transition function $T' = p_\theta^T(s_{i+1}|s_i, a_i; m)$ and an approximate reward function $R' = p_\theta^R(r_{i+1}|s_t, a_t, s_{i+1}; m)$ which are jointly parameterised by $\theta$; and

3. A policy $\pi_\psi(a_t|s_t, q_\phi(m|\tau_{:t}))$ parameterised by $\psi$ and dependent on $\phi$.

The policy is conditioned on both the environment state and the posterior over $m$, $\pi(a_t|s_t, q(m|\tau_{:t}))$. This is similar to the formulation of BAMDPs introduced in 2.2, with the difference that we learn a unifying distribution over MDP embeddings, instead of the transition/reward function directly. This makes learning easier since there are fewer parameters to perform inference over, and we can use data from all tasks to learn the shared reward and transition function. The posterior can be represented by the distribution's parameters (e.g., mean and standard deviation if $q$ is Gaussian).

Our overall objective is to maximise

$$\mathcal{L}(\phi, \theta, \psi) = \mathbb{E}_{p(M)} \left[ \mathcal{J}(\psi, \phi) + \lambda \sum_{t=0}^{H^+} ELBO_t(\phi, \theta) \right]. \tag{10}$$

Expectations are approximated by Monte Carlo samples, and the ELBO can be optimised using the reparameterisation trick (Kingma & Welling, 2014). For $t = 0$, we use the prior $q_\phi(m) = \mathcal{N}(0, I)$. We encode past trajectories using a recurrent network as in Duan et al. (2016); Wang et al. (2016), but other types of encoders could be considered like the ones used in Zaheer et al. (2017); Garnelo et al. (2018); Rakelly et al. (2019). The network architecture is shown in Figure 2.

In Equation (10), we see that the ELBO appears for all possible context lengths $t$. This way, variBAD can learn how to perform inference online (while the agent is interacting with an environment), and decrease its uncertainty over time given more data. In practice, we may subsample a fixed number of ELBO terms (for random time steps $t$) for computational efficiency if $H^+$ is large.

Equation (10) is trained end-to-end, and $\lambda$ weights the supervised model learning objective against the RL loss. This is necessary since parameters $\phi$ are shared between the model and the policy. However, we found that backpropagating the RL loss through the encoder is typically unnecessary in practice. Not doing so also speeds up training considerably, avoids the need to trade off these losses, and prevents interference between gradients of opposing losses. In our experiments, we therefore optimise the policy and the VAE using different optimisers and learning rates. We train the RL agent and the VAE using different data buffers: the policy is only trained with the most recent data since we use on-policy algorithms in our experiments; and for the VAE we maintain a separate, larger buffer of observed trajectories.

At meta-test time, we roll out the policy in randomly sampled test tasks (via forward passes through the encoder and policy) to evaluate performance. The decoder is not used at test time, and no gradient adaptation is done: the policy has learned to act approximately Bayes-optimal during meta-training.

## 4 RELATED WORK

**Meta Reinforcement Learning.** A prominent model-free meta-RL approach is to utilise the dynamics of recurrent networks for fast adaptation ($RL^2$, Wang et al. (2016); Duan et al. (2016)). At every time step, the network gets an auxiliary comprised of the preceding action and reward. This allows learning within a task to happen online, entirely in the dynamics of the recurrent network. If we remove the decoder (Fig 2) and the VAE objective (Eq (7)), variBAD reduces to this setting, i.e., the main differences are that we use a stochastic latent variable (an inductive bias for representing uncertainty) together with a decoder to reconstruct previous *and future* transitions / rewards (which acts as an auxiliary loss (Jaderberg et al., 2017) to encode the task in latent space and deduce information about unseen states). Ortega et al. (2019) provide an in-depth discussion of meta-learning sequential strategies and how to recast memory-based meta-learning within a Bayesian framework.

Another popular approach to meta RL is to learn an initialisation of the model, such that at test time, only a few gradient steps are necessary to achieve good performance (Finn et al., 2017; Nichol & Schulman, 2018). These methods do not directly account for the fact that the initial policy needs to explore, a problem addressed, a.o., by Stadie et al. (2018) (E-MAML) and Rothfuss et al. (2019) (ProMP). In terms of model complexity, MAML and ProMP are relatively lightweight, since they typically consist of a feedforward policy. $RL^2$ and variBAD use recurrent modules, which increases model complexity but allows online adaptation. Other methods that perform gradient adaptation at test time are, e.g., Houthooft et al. (2018) who meta-learn a loss function conditioned on the agent's experience that is used at test time so learn a policy (from scratch); and Sung et al. (2017) who learn a meta-critic that can criticise any actor for any task, and is used at test time to train a policy. Compared to variBAD, these methods usually separate exploration (before gradient adaptation) and exploitation (after gradient adaptation) at test time by design, making them less sample efficient.

**Skill / Task Embeddings.** Learning (variational) task or skill embeddings for meta / transfer reinforcement learning is used in a variety of approaches. Hausman et al. (2018) use approximate variational inference learn an embedding space of skills (with a different lower bound than variBAD). At test time the policy is fixed, and a new embedder is learned that interpolates between already learned skills. Arnekvist et al. (2019) learn a stochastic embedding of optimal $Q$-functions for different skills, and condition the policy on (samples of) this embedding. Adaptation at test time is done in latent space. Co-Reyes et al. (2018) learn a latent space of low-level skills that can be controlled by a higher-level policy, framed within the setting of hierarchical RL. This embedding is learned using a VAE to encode state trajectories and decode states and actions. Zintgraf et al. (2019) learn a deterministic task embedding trained similarly to MAML (Finn et al., 2017). Similar to variBAD, Zhang et al. (2018) use learned dynamics and reward modules to learn a latent representation which the policy conditions on and show that transferring the (fixed) encoder to new environments helps learning. Perez et al. (2018) learn dynamic models with auxiliary latent variables, and use them for model-predictive control. Lan et al. (2019) learn a task embedding with an optimisation procedure similar to MAML, where the encoder is updated at test time, and the policy is fixed. Sæmundsson et al. (2018) explicitly learn an embedding of the environment model, which is subsequently used for model predictive control (and not, like in variBAD, for exploration). In the field of imitation learning, some approaches embed expert demonstrations to represent the task; e.g., Wang et al. (2017) use variational methods and Duan et al. (2017) learn deterministic embeddings.

VariBAD differs from the above methods mainly in what the embedding represents (i.e., task uncertainty) and how it is used: the policy conditions on the posterior *distribution* over MDPs, allowing it to reason about task uncertainty and trade off exploration and exploitation online. Our objective (8) explicitly optimises for Bayes-optimal behaviour. Unlike some of the above methods, we do not use the model at test time, but model-based planning is a natural extension for future work.

**Bayesian Reinforcement Learning.** Bayesian methods for RL can be used to quantify uncertainty to support action-selection, and provide a way to incorporate prior knowledge into the algorithms (see Ghavamzadeh et al. (2015) for a review). A Bayes-optimal policy is one that optimally trades off exploration and exploitation, and thus maximises expected return during learning. While such a policy can in principle be computed using the BAMDP framework, it is hopelessly intractable for all but the smallest tasks. Existing methods are therefore restricted to small and discrete state / action spaces (Asmuth & Littman, 2011; Guez et al., 2012; 2013), or a discrete set of tasks (Brunskill, 2012; Poupart et al., 2006). VariBAD opens a path to tractable approximate Bayes-optimal exploration for

deep RL by leveraging ideas from meta-learning and approximate variational inference, with the only assumption that we can meta-train on a set of related tasks. Existing approximate Bayesian RL methods often require us to define a prior / belief update on the reward / transition function, and rely on (possibly expensive) sample-based planning procedures. Due to the use of deep neural networks however, variBAD lacks the formal guarantees enjoyed by some of the methods mentioned above.

Closely related to our approach is the recent work of Humplik et al. (2019). Like variBAD, they condition the policy on a posterior distribution over the MDP, which is meta-trained using privileged information such as a task description. In comparison, variBAD meta-learns to represent the belief in an unsupervised way, and does not rely on privileged task information during training.

Posterior sampling (Strens, 2000; Osband et al., 2013), which extends Thompson sampling (Thompson, 1933) from bandits to MDPs, estimates a posterior distribution over MDPs (i.e., model and reward functions), in the same spirit as variBAD. This posterior is used to periodically sample a single hypothesis MDP (e.g., at the beginning of an episode), and the policy that is optimal *for the sampled MDP* is followed subsequently. This approach is less efficient than Bayes-optimal behaviour and therefore typically has lower expected return during learning.

A related approach for inter-task transfer of abstract knowledge is to pose policy search with priors as Markov Chain Monte Carlo inference (Wingate et al., 2011). Similarly Guez et al. (2013) propose a Monte Carlo Tree Search based method for Bayesian planning to get a tractable, sample-based method for obtaining approximate Bayes-optimal behaviour. Osband et al. (2018) note that non-Bayesian treatment for decision making can be arbitrarily suboptimal and propose a simple randomised prior based approach for structured exploration. Some recent deep RL methods use stochastic latent variables for structured exploration (Gupta et al., 2018; Rakelly et al., 2019), which gives rise to behaviour similar to posterior sampling. Other ways to use the posterior for exploration are, e.g., certain reward bonuses Kolter & Ng (2009); Sorg et al. (2012) and methods based on optimism in the face of uncertainty (Kearns & Singh, 2002; Brafman & Tennenholtz, 2002). Non-Bayesian methods for exploration are often used in practice, such as other exploration bonuses (e.g., via state-visitation counts) or using uninformed sampling of actions (e.g., $\epsilon$-greedy action selection). Such methods are prone to wasteful exploration that does not help maximise expected reward.

Related to BAMDPs are *contextual MDPs*, where the task description is referred to as a context, on which the environment dynamics and rewards depend (Hallak et al., 2015; Jiang et al., 2017; Dann et al., 2018; Modi & Tewari, 2019). Research in this area is often concerned with developing tight bounds by putting assumptions on the context, such as having a small known number of contexts, or that there is a linear relationship between the contexts and dynamics/rewards. Similarly, the framework of hidden parameter (HiP-) MDPs assumes that there is a set of low-dimensional latent factors which define a family of related dynamical systems (with shared reward structure), similar to the assumption we make in Equation (5) (Doshi-Velez & Konidaris, 2016; Killian et al., 2017; Yao et al., 2018). These methods however don't directly learn Bayes-optimal behaviour but allow for a longer training period in new environments to infer the latents and train the policy.

**Variational Inference and Meta-Learning.** A main difference of variBAD to many existing Bayesian RL methods is that we meta-learn the inference procedure, i.e., how to do a posterior update. Apart from (RL) methods mentioned above, related work in this direction can be found, a.o., in Garnelo et al. (2018); Gordon et al. (2019); Choi et al. (2019). By comparison, variBAD has an inference procedure tailored to the setting of Bayes-optimal RL.

**POMDPs.** Several deep learning approaches to model-free reinforcement learning (Igl et al., 2019) and model learning for planning (Tschiatschek et al., 2018) in partially observable Markov decision processes have recently been proposed and utilise approximate variational inference methods. VariBAD by contrast focuses on BAMDPs (Martin, 1967; Duff & Barto, 2002; Ghavamzadeh et al., 2015), a special case of POMDPs where the transition and reward functions constitute the hidden state and the agent must maintain a belief over them. While in general the hidden state in a POMDP can change at each time-step, in a BAMDP the underlying task, and therefore the hidden state, is fixed per task. We exploit this property by learning an embedding that is *fixed* over time, unlike approaches like Igl et al. (2019) which use filtering to track the changing hidden state. While we utilise the power of deep approximate variational inference, other approaches for BAMDPs often use more accurate but less scalable methods, e.g., Lee et al. (2019) discretise the latent distribution and use Bayesian filtering for the posterior update.

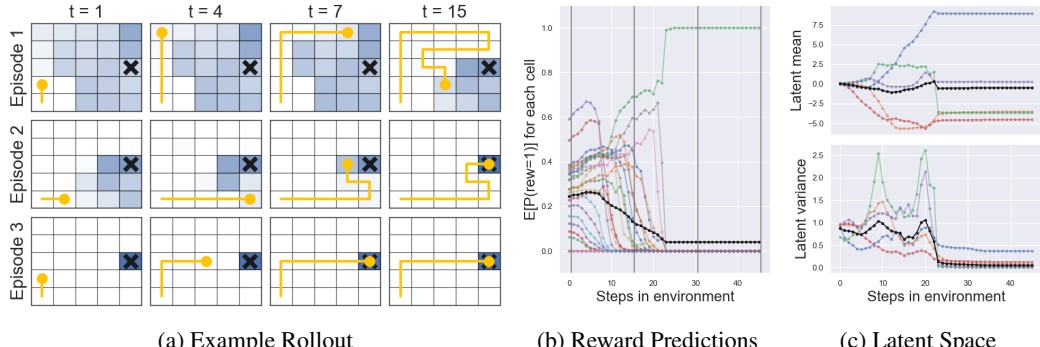

(a) Example Rollout        (b) Reward Predictions        (c) Latent Space

Figure 3: Behaviour of variBAD in the gridworld environment. **(a)** Hand-picked but representative example test rollout. The blue background indicates the posterior probability of receiving a reward at that cell. **(b)** Probability of receiving a reward for each cell, as predicted by the decoder, over the course of interacting with the environment (average in black, goal state in green). **(c)** Visualisation of the latent space; each line is one latent dimension, the black line is the average.

## 5 EXPERIMENTS

In this section we first investigate the properties of variBAD on a didactic gridworld domain. We show that variBAD performs *structured* and *online* exploration as it infers the task at hand. Then we consider more complex meta-learning settings by employing on four MuJoCo continuous control tasks commonly used in the meta-RL literature. We show that variBAD learns to adapt to the task during the first rollout, unlike many existing meta-learning methods. Details and hyperparameters can be found in the appendix, and at `https://github.com/lmzintgraf/varibad`.

### 5.1 GRIDWORLD

To gain insight into variBAD's properties, we start with a didactic gridworld environment. The task is to reach a goal (selected uniformly at random) in a $5 \times 5$ gridworld. The goal is unobserved by the agent, inducing task uncertainty and necessitating exploration. The goal can be anywhere except around the starting cell, which is at the bottom left. Actions are: *up, right, down, left, stay* (executed deterministically), and after $15$ steps the agent is reset. The horizon within the MDP is $H = 15$, but we choose a horizon of $H^+ = 4 \times H = 45$ for the BAMDP. I.e., we train our agent to maximise performance for $4$ MDP episodes. The agent gets a sparse reward signal: $-0.1$ on non-goal cells, and $+1$ on the goal cell. The best strategy is to explore until the goal is found, and stay at the goal or return to it when reset to the initial position. We use a latent dimensionality of $5$.

Figure 3 illustrates how variBAD behaves at test time with deterministic actions (i.e., all exploration is done by the policy). In 3a we see how the agent interacts with the environment, with the blue background visualising the posterior belief by using the learned reward function. VariBAD learns the correct prior and adjusts its belief correctly over time. It predicts no reward for cells it has visited, and explores the remaining cells until it finds the goal.

A nice property of variBAD is that we can gain insight into the agent's belief about the environment by analysing what the decoder predicts, and how the latent space changes while the agent interacts with the environment. Figure 3b show the reward predictions: each line represents a grid cell and its value the probability of receiving a reward at that cell. As the agent gathers more data, more and more cells are excluded ($p(rew = 1) = 0$), until eventually the agent finds the goal. In Figure 3c we visualise the 5-dimensional latent space. We see that once the agent finds the goal, the posterior concentrates: the variance drops close to zero, and the mean settles on a value.

As we showed in Figure 1e, the behaviour of variBAD closely matches that of the Bayes-optimal policy. Recall that the Bayes-optimal policy is the one which optimally trades off exploration and exploitation in an unknown environment, and outperforms posterior sampling. Our results on this gridworld indicate that variBAD is an effective way to approximate Bayes-optimal control, and has the additional benefit of giving insight into the task belief of the policy.

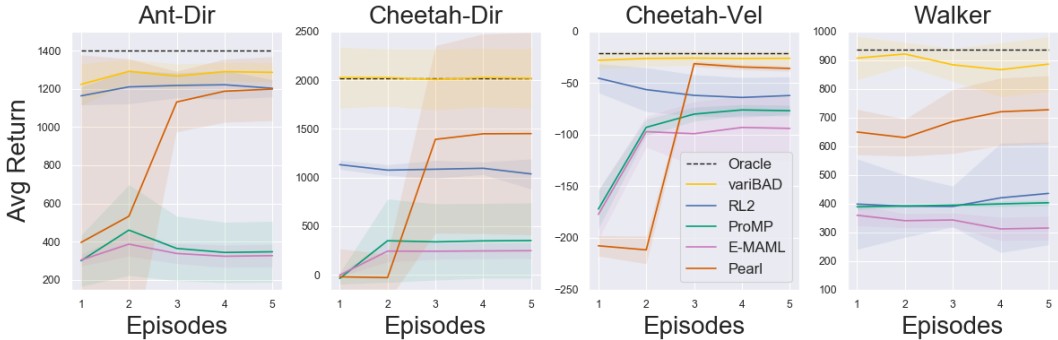

Figure 4: Average test performance for the first 5 rollouts of MuJoCo environments (using 5 seeds).

## 5.2 MuJoco Continuous Control Meta-Learning Tasks

We show that variBAD can scale to more complex meta learning settings by employing it on MuJoCo (Todorov et al., 2012) locomotion tasks commonly used in the meta-RL literature.[2] We consider the AntDir and HalfCheetahDir environment where the agent has to run either forwards or backwards (i.e., there are only two tasks), the HalfCheetahVel environment where the agent has to run at different velocities, and the Walker environment where the system parameters are randomised.

Figure 4 shows the performance at test time compared to existing methods. While we show performance for multiple rollouts for the sake of completeness, anything beyond the first rollout is not directly relevant to our goal, which is to maximise performance on a new task, *while learning, within a single episode*. Only variBAD and RL$^2$ are able to adapt to the task at hand within a single episode. RL$^2$ underperforms variBAD on the HalfCheetahDir environment, and learning is slower and less stable (see learning curves and runtime comparisons in Appendix C). Even though the first rollout includes exploratory steps, this matches the optimal oracle policy (which is conditioned on the true task description) up to a small margin. The other methods (PEARL Rakelly et al. (2019), E-MAML Stadie et al. (2018) and ProMP Rothfuss et al. (2019)) are not designed to maximise reward during a single rollout, and perform poorly in this case. They all require substantially more environment interactions in each new task to achieve good performance. PEARL, which is akin to posterior sampling, only starts performing well starting from the third episode (Note: PEARL outperforms our oracle slightly, likely since our oracle is based on PPO, and PEARL is based on SAC).

Overall, our empirical results confirm that variBAD can scale up to current benchmarks and maximise expected reward within a single episode.

## 6 Conclusion & Future Work

We presented variBAD, a novel deep RL method to approximate Bayes-optimal behaviour, which uses meta-learning to utilise knowledge obtained in related tasks and perform approximate inference in unknown environments. In a didactic gridworld environment, our agent closely matches Bayes-optimal behaviour, and in more challenging MuJoCo tasks, variBAD outperforms existing methods in terms of achieved reward during a single episode. In summary, we believe variBAD opens a path to tractable approximate Bayes-optimal exploration for deep reinforcement learning.

There are several interesting directions of future work based on variBAD. For example, we currently do not use the decoder at test time. One could instead use the decoder for model-predictive planning, or to get a sense for how wrong the predictions are (which might indicate we are out of distribution, and further training is necessary). Another exciting direction for future research is considering settings where the training and test distribution of environments are not the same. Generalising to out-of-distribution tasks poses additional challenges and in particular for variBAD two problems are likely to arise: the inference procedure will be wrong (the prior and/or posterior update) and the policy will not be able to interpret a changed posterior. In this case, further training of both the encoder/decoder might be necessary, together with updates to the policy and/or explicit planning.

---

[2]Environments taken from `https://github.com/katerakelly/oyster`.

ACKNOWLEDGMENTS

We thank Anuj Mahajan who contributed to early work on this topic. We thank Joost van Amersfoort, Andrei Rusu and Dushyant Rao for useful discussions and feedback. Luisa Zintgraf is supported by the Microsoft Research PhD Scholarship Program. Maximilian Igl is supported by the UK EPSRC CDT in Autonomous Intelligent Machines and Systems. Sebastian Schulze is supported by Dyson. This work was supported by a generous equipment grant and a donated DGX-1 from NVIDIA. This project has received funding from the European Research Council under the European Union's Horizon 2020 research and innovation programme (grant agreement number 637713).

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

**Bayes-Adaptive Deep Reinforcement Learning via Meta-Learning**

# Supplementary Material

## A  FULL ELBO DERIVATION

Equation (8) can be derived as follows.

$$
\begin{aligned}
\mathbb{E}_{\rho(M,\tau_{:H})}\left[\log p_\theta(\tau_{:H})\right] &= \mathbb{E}_\rho\left[\log \int p_\theta(\tau_{:H},m)\frac{q_\phi(m|\tau_{:t})}{q_\phi(m|\tau_{:t})}dm\right] \\
&= \mathbb{E}_\rho\left[\log \mathbb{E}_{q_\phi(m|\tau_{:t})}\left[\frac{p_\theta(\tau_{:H},m)}{q_\phi(m|\tau_{:t})}\right]\right] \\
&\geq \mathbb{E}_{\rho,\,q_\phi(m|\tau_{:t})}\left[\log \frac{p_\theta(\tau_{:H},m)}{q_\phi(m|\tau_{:t})}\right] \\
&= \mathbb{E}_{\rho,\,q_\phi(m|\tau_{:t})}\left[\log p_\theta(\tau_{:H}|m)+\log p_\theta(m)-\log q_\phi(m|\tau_{:t})\right] \\
&= \mathbb{E}_\rho\left[\mathbb{E}_{q_\phi(m|\tau_{:t})}\left[\log p_\theta(\tau_{:H}|m)\right]-KL(q_\phi(m|\tau_{:t})||p_\theta(m))\right] \qquad (11)\\
&= ELBO_t.
\end{aligned}
$$

## B  EXPERIMENTS: GRIDWORLD

### B.1  ADDITIONAL REMARKS

Figure 3c visualises how the latent space changes as the agent interacts with the environment. As we can see, the value of the latent dimensions starts around mean 1 and variance 0, which is the prior we chose for the beginning of an episode. Given that the variance increases for a little bit before the agent finds the goal, this prior might not be optimal. A natural extension of variBAD is therefore to also learn the prior to match the task at hand.

### B.2  HYPERPARAMETERS

We used the PyTorch framework for our experiments. Hyperparameters are listed below, and the source code can be found at `https://github.com/lmzintgraf/varibad`.

Hyperparameters for variBAD are:

| | |
|---|---|
| RL Algorithm | A2C |
| Number of policy steps | 60 |
| Number of parallel processes | 16 |
| Epsilon | 1e-5 |
| Discount factor $\gamma$ | 0.95 |
| Max grad norm | 0.5 |
| Value loss coefficient | 0.5 |
| Entropy coefficient | 0.01 |
| GAE parameter tau | 0.95 |
| ELBO loss coefficient | 1.0 |
| Policy LR | 0.001 |
| Policy VAE | 0.001 |
| Task embedding size | 5 |
| Policy architecture | 2 hidden layers, 32 nodes each, TanH activations |
| Encoder architecture | FC layer with 40 nodes, GRU with hidden size 64, output layer with 10 outputs ($\mu$ and $\sigma$), ReLu activations |
| Reward decoder architecture | 2 hidden layers, 32 nodes each, 25 outputs heads, ReLu activations |
| Decoder loss function | Binary cross entropy |

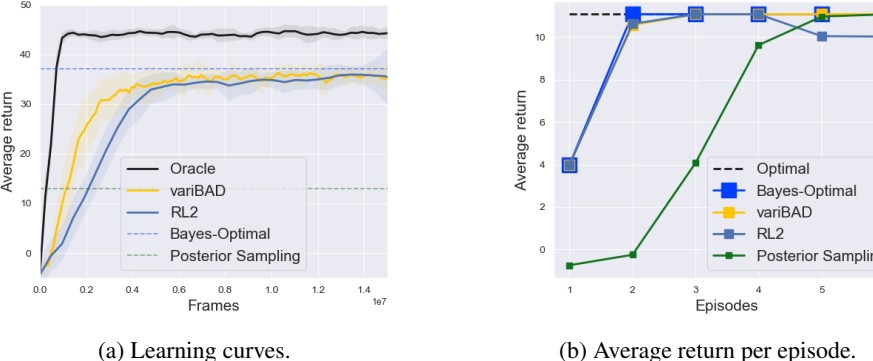

(a) Learning curves.  (b) Average return per episode.

Figure 5: Results for the gridworld toy environment. Results are averages over 20 seeds (with $95\%$ confidence intervals for the learning curve).

Hyperparameters for RL2 the same as above, with the following changes:

| Policy architecture | States are embedded using a fc linear layer, output size 32. Rewards are embedding using a fc layer, output size 8. Results are concatenate and passed to a GRU, hidden size 128, output size 32. After an additional fc layer with hidden size 32, the network outputs the actions. We used TanH activations throughout. |
| --- | --- |

### B.3 COMPARISON TO RL2

Figure 5a shows the learning curves for variBAD and RL2, in comparison to an oracle policy (which has access to the goal position). We trained these policies on a horizon of $H^+ = 4 \times H = 60$, i.e., on a BAMDP in which the agent has to maximise online return within four episodes. We indicate the values of a hard-coded Bayes-optimal policy, and a hard-coded posterior sampling policy using dashed lines.

Figure 5b shows the end-performance of variBAD and RL2, compared to the hard-coded optimal policy (which has access to the goal position), Bayes-optimal policy, and posterior sampling policy. VariBAD and RL2 both closely approximate the Bayes-optimal solution. By inspecting the individual runs, we found that VariBAD learned the Bayes-optimal solution for 4 out of 20 seeds, RL2 zero times. Both otherwise find solutions that are very close to Bayes-optimal, with the difference that during the second rollout, the cells left to search are not all on the shortest path from the starting point.

Note that both variBAD and RL2 were trained on only four episodes, but we evaluate them on six episodes here. After the fourth rollout, we do *not* fix the latent / hidden state, but continue rolling out the policy as before. As we can see, the performance of RL2 drops again after the fourth episode: this is likely due to instabilities in the 128-dimensional hidden state. VariBAD's latent representation, the approximate task posterior, is concentrated and does not change with more data.

## C EXPERIMENTS: MUJOCO

### C.1 LEARNING CURVES

Figure 6 shows the learning curves for the MuJoCo environments for all approaches. The oracle policy was trained using PPO. PEARL (Rakelly et al., 2019) was trained using the reference implementation provided by the authors. The environments we used are also taken from this implementation. E-MAML (Stadie et al., 2018) and ProMP (Rothfuss et al., 2019) were trained using the reference implementation provided by Rothfuss et al. (2019).

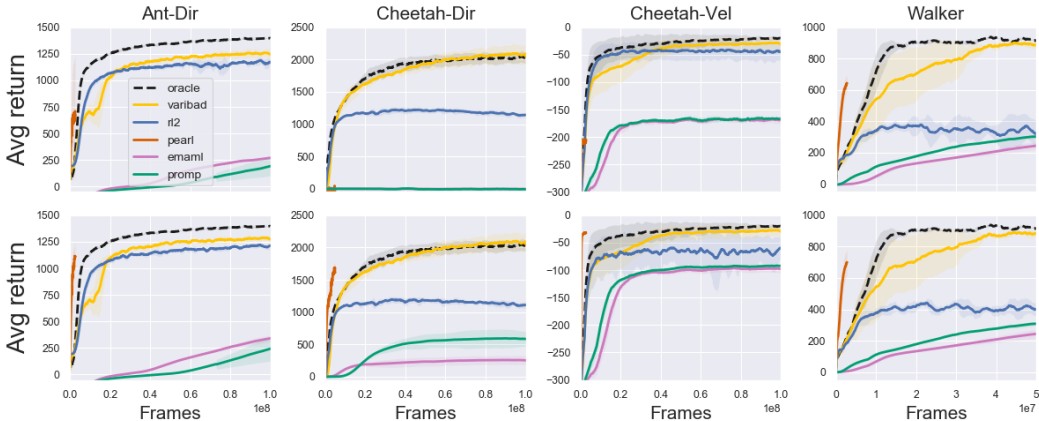

Figure 6: Learning curves for the MuJoCo results presented in Section 5.2. The top row shows performance evaluated at the first rollout, and the second row shows the performance at the $N$-th rollout. For variBAD and RL2, $N = 2$. For ProMP and E-MAML, $N = 20$. For PEARL, $N = 10$.

As we can see, PEARL is much more sample efficient in terms of number of frames than the other methods (Fig 6), which is because it is an off-policy method. On-policy vs off-policy training is an orthogonal issue to our contribution, but an extension of variBAD to off-policy methods is an interesting direction for future work. Doing posterior sampling using off-policy methods also requires PEARL to use a different encoder (to maintain order invariance of the sampled trajectories) which is non-recurrent (and hence faster to train, see next section) but restrictive since it assumes independence between individual transitions.

Note than in Figure 4, for the Walker environment evaluation, we used the models obtained after half the training time ($5e + 7$ frames) for variBAD and the Oracle, since performance declined again after that.

For all MuJoCo environments, we trained variBAD with a reward decoder only (even for Walker, where the dynamics change, we found that this has superior performance).

## C.2 TRAINING DETAILS AND COMPARISON TO RL2

We are interested in maximising performance within a single rollout ($H = 200$). However in order to compare better to existing methods, we trained variBAD and the RL2 baseline to maximise performance within two rollouts ($H^+ = 400$). We implemented task resets by adding a 'done' flag to the states, so that the agent knows when it gets reset in-between episodes. This allows us to evaluate on multiple rollouts (without re-setting the hidden states of the RNN) because the agents have learned to handle re-sets to the starting position.

We observe that RL2 is sometimes unstable when it comes to maintaining its performance over multiple rollouts, e.g., in the CheetahVel task (Figure 6). We hypothesise that the drop of RL2's performance in CheetahVel occurs because it has not properly learned to deal with environment re-sets. The sudden change in state space (with includes joint positions and velocities) could lead to a dramatic shift in the hidden state, which then might not represent the task at hand properly. In addition, once the Cheetah is running at the correct velocity, it can infer which task it is in *from its own velocity* (which is part of the environment state) and stop doing inference, which might be another reason we observe this drop when the environment resets and the state suddenly has a different (very low) velocity. For variBAD this is less of a problem, since we train the latent embedding to represent the task, and only the task. Therefore, the agent does not have to do the inference procedure again when reset to the starting position, but can rely on the latent task description that is given by the approximate posterior. It might also just be due to implementation details, and, e.g., Mishra et al. (2017) do not observe this problem (see their Fig 4).

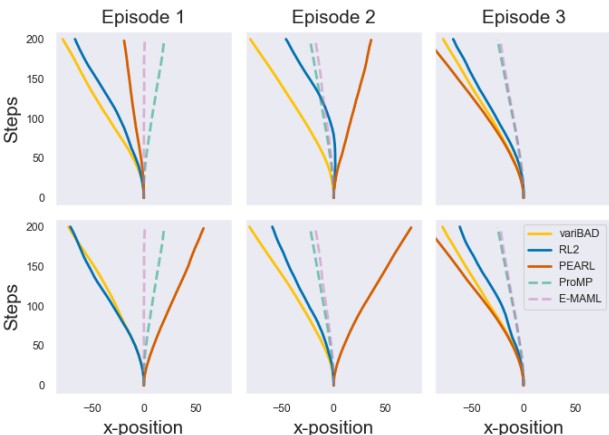

Figure 7: Behaviour at test time for the for the task "walk left" in HalfCheetahDir. The x-axis reflects the position of the agent; the y-axis the steps in the environment (to be read from bottom to top). Rows are separate examples, columns the number of rollouts.

## C.3 CHEETAHDIR TEST TIME BEHAVIOUR

To get a sense for where these differences between the different approaches might stem from, consider Figure 7 which shows example behaviour of the policies during the first three rollouts in the HalfCheetahDir environment, when the task is "go left". Both variBAD and $RL^2$ adapt to the task online, whereas PEARL acts according to the current sample, which in the first two rollouts can mean walking in the wrong direction. For a visualisation of the variBAD latent space at test time for this environment see Appendix C.5. While we outperform at meta-test time, PEARL is more sample efficient *during meta-training* (see Fig 6), since it is an off-policy method. Extending variBAD to off-policy methods is an interesting but orthogonal direction for future work.

## C.4 RUNTIME COMPARISON

The following are rough estimates of average run-times for the HalfCheetah-Dir environment (from what we have experienced; we often ran multiple experiments per machine, so some of these might be overestimated and should be mostly understood as giving a relative sense of ordering).

- ProMP, E-MAML: 5-8 hours
- variBAD: 48 hours
- $RL^2$: 60 hours
- PEARL: 24 hours

E-MAML and ProMP have the advantage that they do not have a recurrent part such as variBAD or $RL^2$. Forward and backward passes through recurrent networks can be slow, especially with large horizons.

Even though both variBAD and $RL^2$ use recurrent modules, we observed that variBAD is faster when training the policy with PPO. This is because we do not backpropagate the RL-loss through the recurrent part, which allows us to make the PPO mini-batch updates without having to re-compute the embeddings (so it saves us a lot of forward/backward passes through the recurrent model). This difference is less pronounced with other RL methods that do not rely on this many forward/backward passes per policy update.

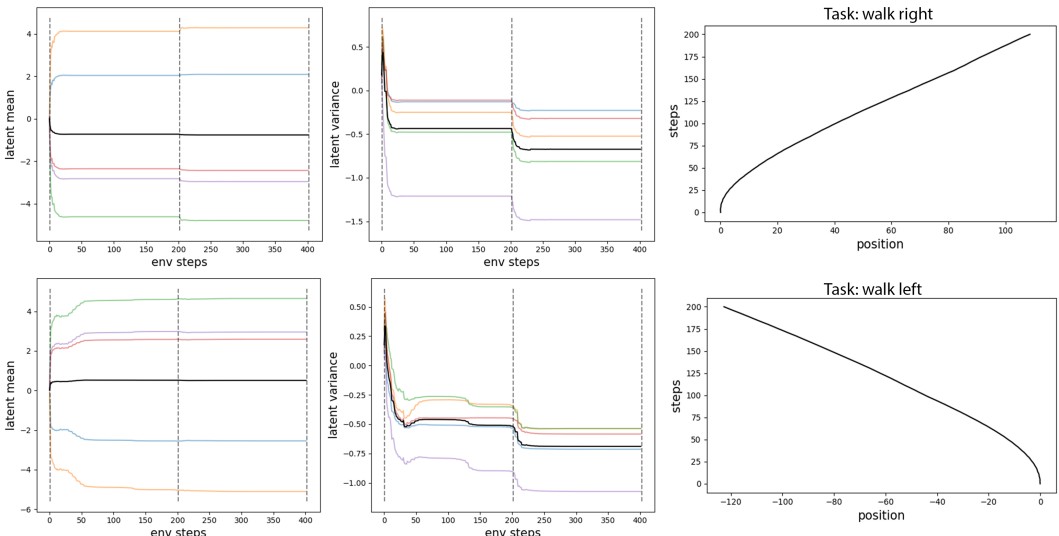

Figure 8: Visualisation of the latent space at meta-test time, for the HalfCheetahDir environment and the tasks "go right" (top) and the task "go left" (bottom). **Left:** value of the posterior mean during a single rollout (200 environment steps). The black line is the average value. **Middle:** value of the posterior log-variance during a single rollout. **Right:** Behaviour of the policy during a single rollout. The x-axis show the position of the Cheetah, and the y-axis the step (should be read from bottom to top).

### C.5    LATENT SPACE VISUALISATION

A nice feature of variBAD is that it can give us insight into the uncertainty of the agent about what task it is in. Figure 8 shows the latent space for the HalfCheetahDir tasks "go right" (top row) and "go left" (bottom row). We observe that the latent mean and log-variance adapt rapidly, within just a few environment steps (left and middle figures). This is also how fast the agent adapts to the current task (right figures). As expected, the variance decreases over time as the agent gets more certain. It is interesting to note that the values of the latent dimensions swap signs between the two tasks.

Visualising the belief in the reward/state space directly, as we have done in the gridworld example, is more difficult for MuJoCo tasks, since we now have continuous states and actions. What we could do instead, is to additionally train a model that predicts a ground-truth task description (separate from the main objective and just for further analysis, since we do not want to use this privileged information for meta-training). This would give us a more direct sense of what task it thinks it is in.

## C.6 HYPERPARAMETERS

We used the PyTorch framework (Paszke et al., 2017) for our experiments. The default arguments for our MuJoCo experiments can be found below, for details see our reference implementation at `https://github.com/lmzintgraf/varibad`.

| | |
|---|---|
| RL Algorithm | PPO |
| Batch size | 3200 |
| Epochs | 2 |
| Minibatches | 4 |
| Max grad norm | 0.5 |
| Clip parameter | 0.1 |
| Value loss coefficient | 0.5 |
| Entropy coefficient | 0.01 |
| Notes | We use a Huber loss in the RL loss |
| Weight of KL term in ELBO | 0.1 |
| Policy LR | 0.0007 |
| Policy VAE | 0.001 |
| Task embedding size | 5 |
| Policy architecture | 2 hidden layers, 128 nodes each, TanH activations |
| Encoder architecture | States, actions, rewards encoder: FC layer (32/16/16-dim), GRU with hidden size 128, output layer with 5 outputs, ReLu activations |
| Reward decoder architecture | 2 hidden layers, 64 and 32 nodes, ReLu activations |
| Reward decoder loss function | Mean squared error |

