# OpenReview forum: "VariBAD: A Very Good Method for Bayes-Adaptive Deep RL via Meta-Learning"
_ICLR.cc/2020/Conference — Accept (Poster)_

### Official Review · AnonReviewer1 · 2019-10-19
**Official Blind Review #1**

**Rating:** 1

**Review:**

The proposed method represents a single MDP using a learned, low-dimensional stochastic latent variable. On these grounds, given a set of tasks sampled from a distribution, the method jointly trains: (1) a variational auto-encoder that can infer the posterior distribution over the postulated latent variable when it encounters a new task while interacting with the environment, and (2) a policy that conditions on this posterior distribution over the MDP embeddings, and thus learns how to trade off exploration and exploitation when selecting actions.

Such variational inference arguments for transfer learning in the context of MDPs are not new. The authors have not made a good job reviewing the related literature. Most importantly, their experimental evaluations lack substantial comparison to such related methods. This is totally disappointing.

**Experience Assessment:**

I have published in this field for several years.

**Review Assessment: Checking Correctness Of Derivations And Theory:**

I carefully checked the derivations and theory.

**Review Assessment: Checking Correctness Of Experiments:**

I carefully checked the experiments.

**Review Assessment: Thoroughness In Paper Reading:**

I read the paper thoroughly.

---

> ### Public Comment · ~Frans_Oliehoek1 · 2019-11-07
> **Criticisms not backed up**
>
> I think this review does not meet the quality requirements.
>
> It makes claims of lack of novelty without backing these up. This is not acceptable for a review. The reviewer should have provided explicit pointers to missed papers, and specify what related
> methods should be compared to.
>
> Therefore, I think this review should be ignored.

---

> ### Author Response · Authors · 2019-11-08
> **Author Reply**
>
> We welcome constructive and fair feedback, positive or negative. Unfortunately, we cannot respond to any of the criticism in this review, since none of the related work which is supposedly missing is named.
>
> Variational inference methods are indeed used in various ways for transfer learning (we give a detailed overview of work in this space on page 7), but many of these settings don’t directly consider the problem of exploration in new tasks. The novelty of our work is that we use VI methods to meta-learn approximately Bayes-optimal policies. We feel like this review disregards large parts of our contribution, discussion of related work, and experimental comparison.
>
> We will be requesting the AC to disregard this unconstructive review.

---

### Official Review · AnonReviewer3 · 2019-10-22
**Official Blind Review #3**

**Rating:** 8

**Review:**

This paper presents a new deep reinforcement learning method that can efficiently trade-off exploration and exploitation. An optimal policy for this trade-off can be solved under the Bayesian-adaptive MDP framework, but in practice, the computation is often intractable. To solve the challenge and approximate a Bayesian-optimal policy, the proposed method VariBAD combines meta-learning, variational inference, and bayesian RL. Specifically, the algorithm learns latent representations of task embeddings and performs tractable approximate inference by optimizing a tractable lower bound of the objective.

The paper is well-written and easy to follow. The combination of meta-learning, variational inference and BAMDP is a clear and neat way to approximate Bayes-optimal policy. The idea also sounds practical for RL as it can approximately solve larger tasks with unknown priors. Experiments on Gridworld and Mujoco show the effectiveness of the proposed method. On Gridworld the performance of the proposed algorithm is close to the performance of the Bayes-optimal policy.

One concern for this paper is the level of novelty, as each major component of the proposed solution has been explored quite extensively in the existing literature (as mentioned in the related work section).

In addition, since comparing many existing Bayesian RL methods, VariBAD meta-learns the inference procedure. This can add additional computation complexity to Bayesian RL, which is not explained or mentioned in neither the method part nor the experiment. I hope the authors can add some discussions on this aspects

**Experience Assessment:**

I have read many papers in this area.

**Review Assessment: Checking Correctness Of Derivations And Theory:**

I assessed the sensibility of the derivations and theory.

**Review Assessment: Checking Correctness Of Experiments:**

I carefully checked the experiments.

**Review Assessment: Thoroughness In Paper Reading:**

I read the paper at least twice and used my best judgement in assessing the paper.

---

> ### Author Response · Authors · 2019-11-08
> **Author Reply**
>
>
>
> Thank you for your review and valuable suggestions. We reply to your points below.
>
> [Novelty]
>
> Our method is one of the few that manages to scale up learning (approximate) Bayes-optimal behaviour to complex environments such as MuJoCo. Unlike existing works, we do not rely on privileged information (such as the task description or ID), and we do not use samples from the posterior (conditioning the policy on the entire posterior like in variBAD makes learning the agent’s strategy harder, since it has to implicitly do planning in belief space, but can ultimately lead to superior performance).
>
> The current state of the art algorithm on the MuJoCo benchmark, PEARL, is akin to posterior sampling, which is not Bayes optimal.  We believe that, by approximating Bayes-optimal exploration at meta-test time, variBAD takes a significant step forward. This is confirmed empirically by variBAD’s, better test-time exploration behaviour and higher performance in the first rollout.
>
> Though we build on concepts that have been explored extensively, variBAD was not straightforward to devise. The exact choice of objective (predicting the future and using the previous posterior as a prior) was crucial in order to get the approximately Bayes-optimal behaviour we want. In that sense, there’s insight in the proposed approach that we think is useful.
>
> [Computational Complexity]
>
> Thank you for pointing this out. We added a discussion of this to the paper (in Appendix C.2, and briefly in the related work section). Here’s a summary:
>
> Indeed, we assume we can meta-train (both the inference procedure and policy) on a set of related tasks, and this is typically computationally expensive: the policy essentially has to learn many tasks at once (which causes meta-RL algorithms to generally take long to learn) and we have additional model complexity due to the encoder/decoder. This setup allows us to save sample costs at test time, which is a desirable property in many situations.
>
> Other existing approximate Bayesian RL methods often rely on sample-based planning (e.g. the work by Arthur Guez), which might include expensive planning steps, and require us to define a prior / belief update on the environment dynamics (which is, e.g., unclear how to do for domains like MuJoCo).
>
> When comparing existing meta-learning methods in terms of runtime, E-MAML and ProMP are fastest. They have the advantage that they do not have a recurrent part such as variBAD or RL^2. Forward and backward passes through recurrent networks can be slow, especially with large horizons. On the other hand, they allow us to do adaptation online, while interacting with the environment.
>
> Even though both variBAD and RL^2 use recurrent modules, we observed that variBAD is faster when training the policy with PPO. This is because we do not backpropagate the RL-loss through the recurrent part, which allows us to make the PPO mini-batch updates without having to re-compute the embeddings (so it saves us a lot of forward/backward passes through the recurrent model). This difference should be less pronounced with other RL methods that do not rely on this many forward/backward passes per policy update.
>
> Compared to PEARL, variBAD takes roughly twice as long to train, which is mostly due to variBAD being on-policy whereas PEARL is off-policy (see figures in the Appendix), but on-policy vs off-policy training is an orthogonal issue to our contribution. Doing posterior sampling using off-policy methods also requires PEARL to use a different encoder (to maintain order invariance of the sampled trajectories) which is non-recurrent (and hence faster to train) but restrictive since it assumes independence between individual transitions.
>
> Please let us know if you have any other questions or concerns.

---

### Official Review · AnonReviewer2 · 2019-10-26
**Official Blind Review #2**

**Rating:** 6

**Review:**

Summary:
This paper considers a version of reinforcement learning problem where an unknown prior distribution over Markov decision processes are assumed and the learner can sample from it. After sampling a MDP, a standard reinforcement learning is done. Then the paper investigates the Bayes-optimal strategy for such meta-learning setting. The experiments are done for an artificial maze solving tasks.

Comments:
Considering a Bayesian setting of reinforcement learning is sound and well-motivated in a mathematical or statistical sense. On the other hand, I wonder what kind of practical applications motivate such formulation. Unfortunately, I don’t have any examples in mind and the paper also shows some artificial experiments. So, the formulation seems, so far, not to be convincing in a practical sense.

Another concern in my mind is that the proposed methods are not supported by any theoretical analyses. I think mathematical papers without practical applications are acceptable if they contain strong mathematical analyses. The present paper, however, does not contain such analyses.

As a summary, I feel that the paper is not strong for theoretical analyses nor practical usefulness, and thus further investigation for either side is necessary.


Comments after Rebuttal:
I modified my score according to authors' comments.

**Experience Assessment:**

I do not know much about this area.

**Review Assessment: Checking Correctness Of Derivations And Theory:**

I assessed the sensibility of the derivations and theory.

**Review Assessment: Checking Correctness Of Experiments:**

I assessed the sensibility of the experiments.

**Review Assessment: Thoroughness In Paper Reading:**

I made a quick assessment of this paper.

---

> ### Author Response · Authors · 2019-11-08
> **Author Reply**
>
>
>
> Thank you for your review. We reply to your points below.
>
> [Motivation]
>
> The scope of applications of our method is huge, since most of RL requires smart exploration. The only assumption that we make is that the agent has the chance to meta-train on a set of related tasks, an assumption made by all of meta-RL. This applies to many settings including, e.g., video games and sim2real transfer for robotics. Our method outperforms the current state of the art meta-learning method (PEARL) on a popular MuJoCo benchmark, in terms of adapting within a single episode. Thus we believe it is a considerable step towards better exploration for RL algorithms via meta-learning.
>
> In many real-world settings, we care not only about performance but we also want our agent to be robust, fair, and safe. Examples are high-stake applications such as healthcare where we care a lot about patient well-being, and education where, e.g., automated tutoring systems should neither bore nor discourage students (for some examples see [1]-[5]). Applying RL to real-world applications like these requires efficient and safe data gathering, i.e., smart exploration.
>
> We added a short motivating sentence to the introduction of the revised version of our paper.
>
> [Theoretical Analysis]
>
> Our method is derived from the problem formulation to approximate the Bayes-optimal solution, which gives it a strong theoretical foundation and motivation. The contribution of our paper is to find scalable approximations to BAMDP solutions, which unsurprisingly precludes theoretical guarantees. We do provide intuition about our objective function by discussing its properties in the paper, and we designed the gridworld experiment to showcase what behaviour we expect and achieve.
>
> Please let us know if you have any other questions or concerns.
>
> [1] Erraqabi, Akram, et al. "Rewards and errors in multi-arm bandits for interactive education." 2016.
> [2] Liu, Yun-En, et al. "Trading Off Scientific Knowledge and User Learning with Multi-Armed Bandits." EDM. 2014.
> [3] Koedinger, Kenneth R., et al. "New potentials for data-driven intelligent tutoring system development and optimization." AI Magazine 34.3 (2013): 27-41.
> [4] Yauney, Gregory, and Pratik Shah. "Reinforcement learning with action-derived rewards for chemotherapy and clinical trial dosing regimen selection." Machine Learning for Healthcare Conference. 2018.
> [5] Hochberg, Irit, et al. "A reinforcement learning system to encourage physical activity in diabetes patients." arXiv preprint arXiv:1605.04070 (2016).

---

> > ### Author Response · Authors · 2019-11-15
> > **Author Reply II**
> >
> >
> >
> > Dear reviewer,
> >
> > We believe we have addressed all of your concerns in our response above and were wondering if you had the chance to look at it. We would appreciate it if you could reconsider your evaluation and score of our paper, or let us know if you have any other questions at this point.

---

### Official Review · AnonReviewer4 · 2019-11-10
**Official Blind Review #4**

**Rating:** 8

**Review:**

*EDIT: Score increased after discussion with authors clarified many concerns raised below*

Summary:
This paper presents an algorithmic approach toward learning Bayes Optimal policies under the uncertainty of the environment. Leveraging meta-learning, the proposed variBAD approximate inference procedure is capable of adapting within the first episode at, what the authors term, meta-test time.

Comments:
It is my estimation that this paper is well positioned to further current state-of-the-art adaptive RL frameworks or methodologies, whether they are meta-learned, transferred or directly inferred through probabilistic mechanisms. The primary contribution of this paper is in how variBAD learns the variational inference procedure. As noted in the related work section, many contemporary policy learning approaches via variational inference are limited by their construction, selection of prior distributions, etc. The advantage of the proposed methodology is that it is capable of efficiently inferring the current environment and adapting the policy learning procedure accordingly. The experiments successfully compare with relevant baselines and prior approaches. The discussion is well framed in highlighting the benefits and limitations of the proposed methodology in relief to prior approaches. One possible weakness in the experimentation, given how closely RL^2 matches the performance of variBAD, is that the specific contributions of each architectural choice or optimization protocol are unclear. There are a few areas in the paper where the authors suggest that ablating their model in specific ways would recover the core approaches present in RL^2. It would be instructive to see how/if performance degrades or converges toward that of RL^2 as the variBAD methodology is ablated.

The paper is well grounded in the literature, albeit skewed perhaps a bit too far toward recent meta-learning results. This is understandable given the focus of this paper, however there are other approaches that might deserve a mention as they similarly parameterize variation over possible MDPs with some latent variables. Namely, I have in mind a few lines of research such as Contextual MDPs (Hallak, et al, 2015; Jiang, et al, 2017; Dann, et al, 2018; etc.), Successor Features (Barretto, et al, 2016,2017,2019; Lehnert, et al, 2017; etc.)  and HiP-MDPs (Doshi-Velez and Konidaris, 2016; Killian, et al, 2017). In particular use of the HiP-MDP framework, Yao, et al (2018) also use the inferred latent variable used to identify the task to condition the policy. While it's always easy to dig into the rabbit holes of related research and distract from the overall objective of a paper, I thought that there was sufficient overlap with these other lines of research that the authors may find interesting. I do not claim that any one of these additional sources of prior work have been overlooked to the detriment of the current paper, they are offered as merely a suggestion to broaden the author's anchoring in the literature.

Now, some more specific questions about the paper and proposed approach. Further clarity along any of these questions would greatly improve the presentation of the paper as well as further convince me of the paper's suitability for publication.
1) What is the advantage of decoding the entire trajectory? It is well understood that this is advantageous in training as that data is available and allows for better inference of the variational parameters. However, under test conditions where the framework may be operating in environments that lie outside the distribution of MDPs it was trained on, I can imagine that errors in trajectory prediction may compound and throw off the entire inference procedure. The experimental set-up did not allay these concerns as there was no mention for holding out-of-distribution tasks/environments aside and the variation in environments is pretty narrow.
2) How is the proposed trajectory decoding more stable than model predictive control? Is stability a large consideration for variBAD when exploring? How much can one trust the exploratory actions under variBAD?
3) The visualization and careful explanation in Section 5.1 of how variBAD executes inference and learning was greatly appreciated. However, are these intuitions valid when extending beyond discrete state and action spaces? Can one make the same claims about the overall approach or procedure in the MuJoCo domains? It was mildly disappointing that a similar explanatory effort was not made in more complex environments. Even an acknowledgement of this being unreasonable would help round out the discussion in Section 5.2.
4) It is not clear what the connection is between the horizon H and the number of rollouts used for evaluation/inference/training. I spent a bit more time than necessary going over and over these items in the paper to where I think that I may understand but I'm still not 100% confident about what is impacted by the number of rollouts used.


**Experience Assessment:**

I have published one or two papers in this area.

**Review Assessment: Checking Correctness Of Derivations And Theory:**

I assessed the sensibility of the derivations and theory.

**Review Assessment: Checking Correctness Of Experiments:**

I carefully checked the experiments.

**Review Assessment: Thoroughness In Paper Reading:**

I read the paper thoroughly.

---

> ### Author Response · Authors · 2019-11-14
> **Author Reply**
>
> Thank you for your thoughtful review and questions, we very much appreciate the time you took to review our work. We reply to your points below.
>
> 1)
>
> The decoder is not used at test time; we only roll out the policy (via forward passes through the encoder and the policy network) without any explicit planning. Instead, the policy has learned to act approximately Bayes-optimally during meta-training. Using the decoder to plan is an interesting direction of future work, but is not trivial for the reasons you mentioned (amongst others). We added some clarification to the latest revision of the paper (end of Sec 3.2).
>
> That being said, we indeed only consider meta-learning settings where the training and test distribution are the same, as common in recent meta-RL literature. We believe however that generalising how to explore in new tasks, even from the same distribution, is already a significant accomplishment.
>
> Transferring to out-of-distribution tasks is even more challenging, and in particular for variBAD two problems are likely to arise: the inference procedure will be wrong (the prior and/or posterior update) and the policy will not be able to interpret a changed posterior. In this case, further training of both the encoder/decoder might be necessary, together with updates to the policy and/or explicit planning. While this is outside the scope of our paper at this point, this is an interesting direction for future research!
>
> 2)
>
> We’re not entirely sure we understand your question correctly. As mentioned above, we do not use the reward/transition model at test time, and do not perform any explicit planning. We believe it is an advantage to not have to do model predictive control, since indeed we do not run into stability problems as you mention. The exploratory actions are deterministically chosen by the policy, and determined by what it has learned to be Bayes-optimal behaviour from meta-training. We hope this answers your question.
>
> 3)
>
> We added a visualisation of the latent space for two sample rollouts in the HalfCheetahDir (tasks: left/right) to Appendix C.3. Visualising the latent space gives us some insight into how fast the posterior concentrates, and in this example we can see this happening within just the first few environment steps.
>
> Visualising the belief in the reward/state space directly is indeed more difficult, since we now have continuous states and actions. What we could do instead, is additionally train a model that predicts a ground-truth task description or ID (separate from the main objective and just for further analysis, since we do not want to use this privileged information for meta-training). This would give us a sense of how certain the agent is about the task (without artefacts such as increasing latent variance in the logspace as we observe in Appendix C.3). We added a note on this to the Appendix as well, and plan to include such visualisations in future revisions of our paper.
>
> 4)
>
> Thank you for pointing this out! We tried to clarify this in the paper (by renaming H in the BAMDP to H^+, and an explanation in section 2.2 shortly after Eq (3)).
>
> Bayes-Optimal behaviour critically depends on the number of time steps given to the agent. I.e., optimally trading off exploration and exploitation can lead to very different behaviour when the agent is given only 10 steps, vs. when it is given 100 steps. E.g. let’s say the agent can learn something about the task at hand without getting any reward (purely information-seeking actions), then these might be worth it only if there is enough time to exploit that information. If, however, time’s up after learning that information, a better strategy is to take a gamble and try to solve the most likely task directly.
>
> Therefore, when training variBAD, we need to pre-specify for which horizon we want the agent to act Bayes-optimal. In the gridworld, this was three episodes of the original MDP (so H^+=3*15, which is the new horizon in the BAMDP), and in MuJoCo this was one episode (so H^+=1*200).
>
> Other)
>
> Thank you for pointing out the additional literature, this is very much appreciated. We will broaden our review of related work in the paper as we dive into these.

---

> > ### Comment · AnonReviewer4 · 2019-11-14
> > **Thank you**
> >
> > Thank you for the additional clarification. I feel as though my questions were satisfactorily answered for the most part. I am inclined to improve my initial score. Below are some follow-ups to the authors' responses.
> >
> > Before I respond to these points, I do want to elevate one comment made in my initial review, that of showing ablations to the proposed variBAD approach so as to highlight how performance changes relative to the RL^2 baseline, as discussed in the paper.
> >
> > 1-2) Your responses helped clarify and correct my misunderstanding of your use of the decoder. As the authors noticed, I mistakenly believed that the decoder was used for planning purposes. This eliminates my concerns about compounding model errors and stability. The additional clarification added to the paper was helpful as well.
> >
> > I agree that the contributions made to improve exploration when operating on in-distribution tasks are important. While I respect the advances made possible by meta-RL, I am disappointed that so much effort (computational and intellectual) is being spent fitting to a narrow distribution of tasks without discussing the limitations of such methods. I appreciate the authors' effort to highlight these limitations in this discussion, I would advocate that such points should be included in the paper. Moving to "generalize" to out-of-distribution tasks *is* an important and exciting area of research. There has been some work along these lines (Nagabandi+Clavera, et al [ICLR 2019]), I wonder if the variational inference framework presented in this paper, or a modified version of VIREL (Fellows+Mahajan, et al [NeurIPS 2019]) could help paper over some of the computational inefficiencies and hand tuned model selection.
> >
> > 3) I appreciate the additional efforts taken to provide some consistency in how the authors analyze variBAD between experimental domains. I understand that having a continuous state and action space do complicate a similar demonstration as is used in the grid world experiments.
> >
> > As a point of suggestion for the authors' proposed additional experiments, the PEARL paper constructed tasks in the Ant domain where the agent was expected to navigate to a particular location. Perhaps having a spatial component of the task could help visualize the agent's belief of the goal state in these continuous environments. I'm not sure whether the added complexity with an increased state and action space would complicate the use of variBAD.
> >
> > Additionally, Nagabandi, et al [ICRA 2018] (among others, I'm sure) constructed specific trajectories for their agent to follow. Their success however depended heavily on MPC which may not lend itself directly to the proposed approach.
> >
> > 4) This was helpful, thank you.

---

> > > ### Author Response · Authors · 2019-11-15
> > > **Author Reply**
> > >
> > >
> > >
> > > Thank you for the additional pointers.
> > >
> > > 1-2) We agree with your sentiment regarding generalisation to out-of-distribution task, and hope to see more research towards this in the future. We added a short discussion on future work to our conclusion to touch on these points with the reader.
> > >
> > > 3) Thank you! Yes, the AntGoal environment seems like a good testbed for visualising the belief also for continuous state spaces. In theory, there is nothing that should prevent variBAD from learning with even larger state / action space, but we haven’t yet performed these experiments. We will look into scaling variBAD up even further in the future.

---

### Decision · Program_Chairs · 2019-12-19

**Decision:**

Accept (Poster)

**Comment:**

This paper considers the problem of transfer learning among families of MDP, and proposes a variational Bayesian approach to learn a probabilistic model of a new problem drawn from the same distribution as previous tasks, which is then leveraged during action selection.

After discussion, the three respondent reviewers converged to the opinion that the paper is novel and interesting, and well evaluated. (Reviewer 1 never responded to any questions the authors or me, so I have disregarded their review.) I am therefore recommending an accept.